# Bone-Targeted Agents and Metastasis Prevention

**DOI:** 10.3390/cancers14153640

**Published:** 2022-07-26

**Authors:** Robert Coleman

**Affiliations:** Department of Oncology and Metabolism, University of Sheffield, Sheffield S10 2RX, UK; r.e.coleman@sheffield.ac.uk

**Keywords:** breast cancer, adjuvant therapy, bisphosphonates, denosumab, biomarkers, bone-targeted treatments

## Abstract

**Simple Summary:**

Bone metastases are common in cancer patients and the role of bone-targeted agents in prevention of metastasis have been evaluated in multiple clinical trials over the past 20 years. Results show that in breast cancer the use of adjuvant bisphosphonates (and possibly denosumab) reduce bone metastases and breast cancer deaths in postmenopausal women. These effects are in addition to the benefits associated with the use of standard adjuvant endocrine, cytotoxic and targeted treatments with prevention of one in six breast cancer deaths at 10 years. Similar benefits have not been observed in other cancers that typically spread to bone, such as prostate or lung cancers. Biomarkers that can predict patient benefit from the use of bone targeted treatments in the adjuvant setting are being evaluated. Currently, tumour expression of the transcription factor, MAF, seems to be the most promising biomarker; benefits from adjuvant bisphosphonates are seen in the 80% of patients with normal levels of expression irrespective of menopausal status, while over-expression is associated with a poor prognosis and a higher rate of visceral metastases.

**Abstract:**

The use of bone-targeted treatments has transformed the clinical care of many patients with metastatic breast cancer. In addition, due to the profound effects of bisphosphonates and denosumab on bone physiology and the bone microenvironment, the potential of bone-targeted agents to modify the process of metastasis has been studied extensively. Many adjuvant trials with bisphosphonates in early breast cancer have been performed. Variable outcomes in terms of disease recurrence have been reported, with any treatment benefits apparently influenced by the age and menopausal status of the patients. The Early Breast Cancer Trialists’ Collaborative Group (EBCTCG) conducted a meta-analysis of individual patient data from all available randomised trials to investigate this observation further. This meta-analysis failed to show any benefits of adjuvant bisphosphonates in premenopausal women, but highly significant improvements in bone recurrence (RR = 0.72; 95% CI 0.60–0.86, 2*p* = 0.0002) and breast cancer mortality (RR = 0.82; 95% CI 0.73–0.93, 2*p* = 0.002) were seen in the 11,767 postmenopausal women included in the meta-analysis. As a result, clinical guidelines recommend the incorporation of adjuvant bisphosphonates that inhibit osteoclast activity into routine clinical care. Denosumab, which has similar effects on bone cell physiology, appears not to consistently influence disease outcomes, perhaps suggesting that it is the “off target” effects of bisphosphonates on immune function and the biological processes involved in metastasis that are important. Predictive biomarkers beyond menopause are being sought and assessment of the transcription factor MAF (mesenchymal aponeurotic fibrosarcoma gene) appears to identify patients able to benefit from the addition of a bisphosphonate to standard adjuvant anticancer therapies.

## 1. Introduction

The biology of metastasis in cancer is extraordinarily complex, composed of multiple steps and orchestrated by a range of interactions between the host and both primary and disseminated cancer cells. Exosomes, microRNAs and growth factors released from the primary tumour are thought to initiate the process by preparing the pre-metastatic niche [1]. The primary tumour then releases cells that are able to pass through the extracellular matrix and enter the vasculature, before being transported to distant organs. Only a small proportion of disseminated tumour cells (DTCs) survive this multi-step process and are thus able to colonise potential metastatic sites. The bone marrow microenvironment is thought to provide the favourable conditions required for malignant cells to survive. Once in bone, DTCs find a hospitable environment or niche to enter. Niches for haematopoietic stem cells, osteoblasts and the bone vasculature within the bone microenvironment have all been suggested as favourable sites for DTC to enter [2]. Once established within the microenvironment, DTCs may enter a non-proliferating dormant state, which can persist for many years. The mechanisms by which dormant cells emerge from this quiescent state are not well understood. Multiple cellular interactions, changes in immunological and growth factor control, as well as the physical properties of the microenvironment have all been postulated as potentially important in the development of overt metastases. Disease relapse often occurs many years after the initial diagnosis of the primary tumour, and tumour dormancy is a particular feature of breast and prostate cancers [3].

Tumour cells within the bone microenvironment have the capacity to produce a wide range of cytokines and growth factors as they emerge from the dormant state. These include parathyroid hormone-related peptide (PTHrP), prostaglandins and interleukins that can stimulate osteoblast production of receptor activator of nuclear factor kappa B ligand (RANKL). Activation of osteoclasts occurs, resulting in disruption of the normal balance of new bone formation and bone resorption. Bone-derived growth factors are released from the resorbing bone matrix that, at least in animal models, have been shown to increase proliferation of tumour cells within the bone microenvironment. This cross-talk between host cells and metastatic tumour cells, sometimes referred to as the vicious cycle, facilitates the development of metastases within bone and, potentially, also at extraosseus sites for metastasis [4].

## 2. Role of Bone-Targeted Treatments to Prevent Metastasis

Undoubtedly, the extraordinarily complex biology of metastasis and control of tumour dormancy requires further study. Our current understanding is somewhat speculative and derived from preclinical models that may not accurately reflect the processes involved in metastasis within patients. Nevertheless, treatments that modify bone cell function and change cellular interactions within the bone microenvironment could potentially influence the development of metastases and change the clinical course of the disease. Currently, the bone-targeted agents, such as the bisphosphonates and denosumab, provide the best option to potentially influence the processes involved in metastasis. Numerous animal model systems have yielded promising results [5], and numerous clinical studies have been performed, especially in early breast cancer, as discussed below.

### Adjuvant Treatment of Early Breast Cancer

Systemic treatments with chemotherapy, endocrine agents in oestrogen receptor positive (ER+) breast cancer and targeted therapies such as trastuzumab in patients with tumours over-expressing the HER2 growth factor receptor, so called adjuvant therapy, have greatly improved the prognosis of early (stage I–III) breast cancer. Clinical trials evaluating the potential role of adding bone-targeted treatments in the adjuvant treatment of early breast cancer were initiated more than 25 years ago. Individual studies have been difficult to interpret as varying results were reported. For example, several early studies with daily oral clodronate suggested fewer bone relapses and improved survival [6,7] but contrary results were also reported [8] and, as a result, clodronate was not approved as an adjuvant treatment strategy. The Austrian Breast Cancer Study Group (ABCSG) conducted a potentially practice-changing study in which they reported significant benefit from the addition of six monthly zoledronate to endocrine therapy that included ovarian suppression for premenopausal women with ER+ stage 1 or stage 2 breast cancer [9]. Subsequently, however, the AZURE study, with broader inclusion criteria and testing a more intensive treatment regimen with zoledronate, showed no benefit in an intention to treat (ITT) analysis, either at the time of the initial analysis [10] or at later planned follow-up analyses [11,12]. Nevertheless, the AZURE study did identify improved outcomes in the subgroup of patients who were postmenopausal at the time of study entry [10]. This finding, along with the benefits seen in ABCSG-12 and several bone protection studies conducted in postmenopausal women, generated the hypothesis that treatment benefits were restricted to women who had low levels of reproductive hormones, either due to a natural age-related menopause or secondary to ovarian function suppression [13].

The Early Breast Cancer Trialists’ Collaborative Group (EBCTCG) investigated this hypothesis by performing a meta-analysis of individual patient data collected from 18,766 patients with early breast cancer who had been included in randomised trials of adjuvant bisphosphonates. The meta-analysis showed that adjuvant bisphosphonates (any of intravenous zoledronate, daily oral clodronate or daily oral ibandronate) reduced breast cancer recurrences (especially in bone) and breast cancer deaths. However, benefit from the addition of adjuvant bisphosphonates to standard adjuvant treatment was restricted to women who were postmenopausal at the time treatment was initiated [14]. Across all age and menopausal groups, the adjuvant use of a bisphosphonate had no significant effect on the rates of overall recurrence (rate ratio [RR] = 0.94, 95% CI 0.87–1.101, 2*p* = 0.08) despite a reduction in the frequency of bone metastases. Additionally, the effect on breast cancer mortality, although statistically significant, was small in absolute terms (RR = 0.91, 95% CI 0.83–0.99, 2*p* = 0.04). However, clinically important benefits were seen in postmenopausal women or those receiving ovarian function suppression, confirming the hypothesis proposed by the AZURE investigators. In this population of 11,767 women, a large reduction in bone recurrence (RR = 0.72, 95% CI 0.60–0.86, 2*p* = 0.0002) led to statistically and clinically significant improvements in overall breast cancer recurrence (RR = 0.86, 95% CI 0.78–0.94, 2*p* = 0.02) and, most importantly, a clinically important reduction in breast cancer-specific mortality (RR = 0.82, 95% CI 0.73–0.93, 2*p* = 0.002) [14], with prevention of more than one in six breast cancer deaths at 10 years, equivalent to a potential for >10,000 fewer deaths per year from breast cancer across the European Union and United Kingdom.

Treatment benefits were similar across biological subtypes of breast cancer (both ER+ and ER- as well as low-, intermediate- and high-grade tumours) and were seen irrespective of the type of bisphosphonate used. Supporting the notion that the benefits in breast cancer outcomes were a class effect across the different bisphosphonates tested, a large, randomised trial (SWOG0307) showed no differences in disease recurrence or breast cancer deaths between intravenous zoledronate, daily oral clodronate or daily oral ibandronate [15]. However, there is no randomised trial evidence available to say whether relatively low doses of oral bisphosphonates, such as weekly alendronate or risedronate, used widely for the prevention and treatment of osteoporosis, are sufficient to modify the biology of breast cancer [14].

Adjuvant trials with denosumab have also been performed in early breast cancer to see if this bone-targeted treatment also has disease modifying effects. The ABCSG evaluated the osteoporosis schedule of denosumab in study ABCSG 18. This trial was designed primarily to assess whether denosumab could prevent bone loss and reduce the frequency of fractures associated with the use of aromatase inhibitors [16]. In an update of the study, the secondary endpoints of disease-related outcomes were reported [17]. A significant improvement in disease-free survival (DFS) was suggested, with 240 events in the denosumab treated patients versus 287 in the placebo group (HR 0.82, 95% CI 0.69–0.98, *p* = 0.026). However, at this analysis, the reduction in DFS events was due largely to fewer second non-breast primary cancers (80 denosumab versus 100 placebo events) and deaths without recurrence (48 denosumab versus 39 placebo events), rather than a reduction in breast cancer recurrences. Histologically confirmed breast cancer events were similar in the two treatment groups (73 denosumab versus 68 placebo events). Very recently, the long-term follow-up results from this study have been presented and suggest, in this population of relatively good prognosis postmenopausal women receiving aromatase inhibitors, that adjuvant denosumab may be influencing the course of the disease. With a median follow-up of 8 years, patients treated with denosumab not only had a significant improvement in DFS (denosumab 309 versus placebo 368 DFS events, hazard ratio (HR) 0.83, 95% CI 0.71–0.97, *p* = 0.016), but also a potential improvement in overall survival (denosumab 127 versus placebo 158 OS events, HR 0.80, 95% CI 0.64–1.01, *p* = 0.065) [18].

The D-CARE study randomised 4509 women with histologically-confirmed stage II/III breast cancer to receive either subcutaneous denosumab (120 mg) or a matching placebo. Treatment was administered every 4 weeks for 6 months and then continued every 3 months to complete a total duration of study treatment of 5 years [19]. Bone metastasis- free survival (BMFS) was the primary endpoint. This composite endpoint was defined as the time from randomisation to the first observation of bone metastasis, with or without disease recurrence at other anatomical sites or death from any cause. All imaging was reviewed centrally to confirm recurrence. Disease relapse not conclusively shown on imaging had to be confirmed by biopsy. Denosumab had no significant effect on BMFS (hazard ratio [HR] = 0.97, 95% CI 0.82–1.14; *p* value = 0.70). Additionally, there were no benefits in disease-free or overall survival with denosumab, either in the study population as a whole or, unlike in the bisphosphonate trials, in the subgroup of women who were postmenopausal at study entry. Exploratory analyses suggested a reduction in first recurrence in bone and other positive effects on bone-related endpoints [20], but this was balanced by an excess in disease recurrence at extra-skeletal distant metastatic sites. The difference in results between D-CARE and those identified by the EBCTCG meta-analysis may indicate that the benefits of bisphosphonates relate to the broader and more sustained biological effects on multiple aspects of the metastatic process, and not just through their primary effects on bone cell function. Denosumab, in contrast, acts only on cells expressing RANK receptors and influenced by levels of RANK ligand and thus, while denosumab is a potent inhibitor of osteoclast function, it is only able to directly influence a small proportion of breast cancer cells. Challenging this hypothesis is the apparent benefit in patients treated with denosumab in the ABCSG 18 trial [18]. As will be shown below, the benefits of adjuvant bisphosphonates are not only influenced by the menopausal status of the patient but probably also by the biology of the underlying tumour.

## 3. Clinical Application and Treatment Guidelines

Bone-targeted agents have two indications in the context of early breast cancer: firstly, the prevention of treatment-induced bone loss associated with aromatase inhibitors or induction of menopause and, secondly, the inhibition of metastasis. Metastasis prevention should be the primary concern for patients at intermediate to high risk of recurrence. For such patients, zoledronate, typically initiated alongside (neo)adjuvant chemotherapy and then 4 mg every 6 months or daily oral ibandronate or clodronate can be considered (Figure 1).

The optimum duration of adjuvant bisphosphonate treatment in early breast cancer to achieve the reductions seen in bone metastasis and breast cancer deaths is uncertain. The EBCTCG meta-analysis provides indirect evidence for similar outcomes between trials that tested treatment durations of between 2 and 5 years [14]. The only randomised evidence comes from the SUCCESS trial, which compared 2- or 5-year schedules of zoledronate, and would seem to support these observations. Although somewhat underpowered, no differences in either disease outcome or the presence of DTCs in peripheral blood between the two schedules were seen [22].

As an alternative to patient selection based on risk of recurrence and menopausal status, as recommended by guidelines both in Europe [21] and North America [23], efforts are ongoing to identify biomarkers that may select patients able to benefit from manipulation of the bone microenvironment. To date, the most promising biomarker for patient selection is MAF (mesenchymal aponeurotic fibrosarcoma gene), an AP-1 family transcription factor that is amplified in around 20% of primary breast tumours. MAF is a transcription factor that controls many genes, including CD36 and PTHrP, and regulates a diverse range of metastasis-related cellular processes, including initiation of metastasis, adhesion to bone marrow-derived cells and metabolic rewiring, as well as osteoclast differentiation, suggesting that MAF may play a key role in metastasis, especially to bone [24].

MAF was initially shown to be a prognostic factor that predicts for bone recurrence [25] and can be assessed in formalin fixed paraffin embedded sections of the primary tumour using a fluorescent in-situ hybridisation (FISH) assay (MAFTEST™). The clinically more important observation that MAF status could predict benefits (and harm) from the use of adjuvant bisphosphonates in early breast cancer came from an evaluation of 879 patients included in the AZURE trial. With a pre-specified cut-off of 2.5 copies of MAF per centromere denoting MAF positivity, 21% of tumours were classified as MAF positive and 79% were MAF negative. Zoledronate improved invasive disease-free and overall survival in the 80% of patients who were MAF negative, irrespective of menopausal status, both at an initial analysis after 84 months follow-up [26] and at the final follow-up at 10 years after study entry [10]. In contrast, those patients with amplification of MAF and classified as MAF positive appeared to have a worse outcome when treated with zoledronate, with a marked increase in recurrences at extraosseus sites and worse overall survival.

These hypothesis-generating findings have since been validated in an analysis of MAF status using the same FISH assay and cut-off for positivity in a larger population of patients included in a randomised trial of daily oral clodronate in early-stage breast cancer (NSABP-B34) [27]. Just like in the AZURE trial, no significant effects of clodronate on disease outcomes were seen in the ITT analysis of the study as a whole. However, in the 80% of patients with a MAF negative FISH test, significant improvements in both disease-free and overall survival were seen, again irrespective of menopausal status. In contrast, those with MAF positive tumours had no benefit, with a numerically worse outcome [28]. Table 1 shows the results of a recent combined analysis of MAF testing in AZURE and B-34, suggesting that utilising MAF as a new selection criterion not only reduces the risk of death and relapse for selected patients, over and above those identified using the current clinical guidelines [21,22] but also provides a new treatment option for young patients currently excluded from this treatment in clinical practice [29].

It would be intriguing to know the MAF status of patients included in the D-CARE and ABCSG 18 trials, respectively. The relatively poor prognosis population recruited to D-CARE would argue for a high proportion having MAF positive tumours; perhaps the benefit seen in ABCSG 18 relates to the low risk of relapse in the study population and a likely very low rate of MAF positivity. Sadly, evaluation of the tumours collected in D-CARE is no longer possible [30].

Treatment recommendations for the use of bone targeted treatments as disease-modifying agents are summarised in Table 2.

### Adjuvant Bone-Targeted Agents in Other Tumor Types

While adjuvant bisphosphonates for women with early breast cancer have become part of routine clinical practice in many parts of the world, results from trials evaluating bone-targeted agents as disease-modifying agents in other disease settings have been disappointing. Because prostate cancer spreads predominantly to bone, it provides an ideal clinical setting for the evaluation of bone-targeted treatments. However, the benefits seen in breast cancer were not mirrored by trials conducted in this disease setting. For example, the Zometa European Study (ZEUS), which compared zoledronate every 3 months to placebo infusions in patients with non-metastatic prostate cancer deemed at high-risk for developing overt bone metastasis, showed no benefit from this bone-targeted treatment. After a median follow-up of 4.8 years, the proportions developing bone metastases were 14.7% with zoledronate and 13.2% in the placebo arm [31]. Similarly, a meta-analysis of all available randomised data in prostate cancer showed no measurable impact from bisphosphonates on disease recurrence, progression or survival [32].

Denosumab has also been tested as a disease-modifying agent in prostate cancer in men with castrate-resistant prostate cancer (defined as a confirmed rising PSA despite castrate levels of testosterone) but no evidence of overt metastases. Denosumab significantly increased bone metastasis-free survival (the primary endpoint of the trial) by a median of 4.2 months over a placebo (HR = 0.85; *p* = 0.028), and delayed time to symptomatic first bone metastases, although there was no impact on survival [33]. Although the trial met its primary endpoint, the disease benefits were offset by a 5% incidence of osteonecrosis of the jaw. As a result, the benefits were not considered clinically sufficient for regulatory approval.

The potential impact of bone-targeted agents on the natural history of lung cancer has also been evaluated. However, neither zoledronic acid [34] or denosumab [35] had any measurable impact on disease recurrence or survival.

## 4. Future Directions

In addition to development of biomarkers to predict patients most likely to benefit from bone-targeted treatments, novel agents are needed that can modify the bone microenvironment and reduce the potential for metastasis. Radium-223. is an alpha particle- emitting radiopharmaceutical that is approved for the treatment of metastatic castrate-resistant prostate cancer. Studies evaluating its potential use in breast cancer are limited and use in the adjuvant setting has not been examined due, at least in part, to the theoretical concerns about long term carcinogenesis in a population of potentially cured patients. Pre-clinical studies have been promising [36]. and the dosimetry of radium-223, with it preferentially taken up on the endosteal bone surface and irradiating just a few cell diameters from the bone surface where malignant cells are likely to reside in the metastatic niches, make it a potential agent for future study in women with high risk early breast cancer.

## 5. Conclusions

Adjuvant bisphosphonates are recommended as adjuvant treatment in postmenopausal women with early breast cancer deemed at intermediate to high risk of disease recurrence. While denosumab is a highly effective agent for the prevention of fractures associated with treatment-induced bone loss, it does not appear to modify the clinical course of breast cancer. Further research is needed to identify strategies to prevent disease recurrence in premenopausal breast cancer and other diseases associated with the development of bone metastases.

## Figures and Tables

**Figure 1 cancers-14-03640-f001:**
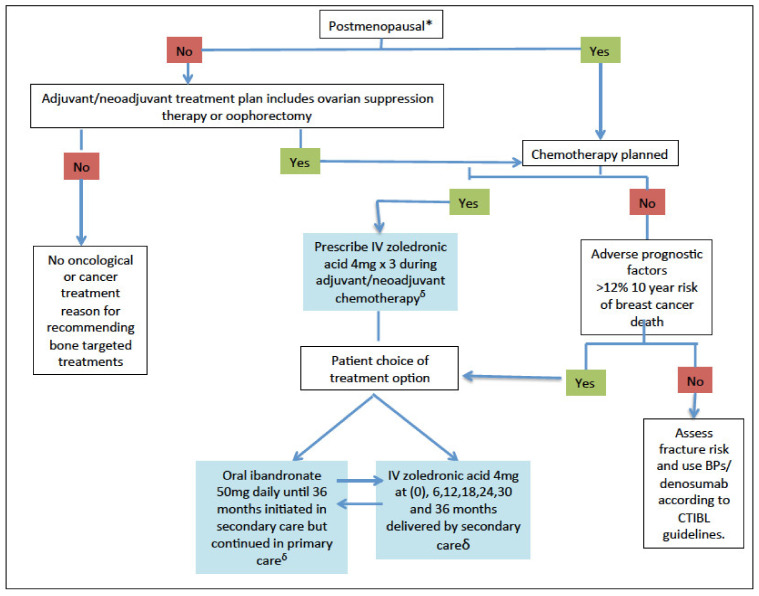
Algorithm for the use of adjuvant bisphosphonates in patients^#^ with early breast cancer. Adapted with permission from ESMO guidelines on bone health from Ref. [21], 2020, Annals of Oncology, Elsevier. * If not clinically assessable, i.e., hysterectomy/IUD, then ensure age > 55+/or serum FSH is in postmenopausal range (patient must not be receiving concurrent therapies that can affect the hypophyseal pituitary gonadal axis). # Patients already on weekly oral bisphosphonates for osteoporosis should be considered for a treatment change and follow algorithm. ^δ^ Include vitamin D 800–2000 IU (+calcium 1000 mg daily if low calcium diet). CTIBL, cancer therapy induced bone loss; IV, intravenous.

**Table 1 cancers-14-03640-t001:** Combined analysis of the impact of adjuvant bisphosphonates on outcomes of patients in AZURE (discovery set [12]) and NSABP B34 trials (validation set [27]) according to MAF testing on the primary tumour.

Patient Subgroup	Number of Patients	IDFS Events with BPs at 10 Years	IDFS Events in Control Group at 10 Years	OS Events with BPs at 10 Years	OS Events in Control Group at 10 Years
**All patients**	2810	351/1356 (25.9%)	376/1392 (27.0%)	211/1356 (15.6%)	254/1392 (18.3%)
**MAF+**	514 (18%)	98/286 (34.3%)	79/266 (29.7%)	71/286 (24.9%)	62/266 (23.3%)
**MAF−**	2296 (82%)	253/1070 (23.6%)	297/1126 (26.4%)	140/1070 (13.1%)	192/1126 (17.1%)

IDFS, invasive disease-free survival; OS, overall survival; BPs, adjuvant bisphosphonates.

**Table 2 cancers-14-03640-t002:** Treatment recommendations for use of bone-targeted agents as disease-modifying agents in cancer patients.

Adjuvant bisphosphonates (intravenous zoledronic acid, or daily oral clodronate or ibandronate) are recommended for postmenopausal women or premenopausal women treated with GnRH analogues with early breast cancer deemed at significant risk for recurrence.
Treatment should be initiated alongside (neo)adjuvant chemotherapy (where indicated) and continued for 2–5 years.
Bisphosphonates are not recommended as disease-modifying agents for premenopausal women (not on GnRH analogues) with early breast cancer, * in men with localised prostate cancer, nor patients with other solid tumours ^#^.
Denosumab is not (currently) recommended for the prevention of metastasis

* The MAFTEST™, which very recently has become commercially available, may provide an option for selecting premenopausal women able to benefit from the addition of adjuvant bisphosphonates. # Excludes multiple myeloma where bone targeted agents appear to extend the time to progression and overall survival.

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
