# Peer review of "Bone-Targeted Agents and Metastasis Prevention"

_cancers, 2022, doi:10.3390/cancers14153640_

Round 1

Reviewer 1 Report

Excellent idea to present an updated and complex  study of the bisphosphonates and their implication as adjuvants in cancer treatment. The design of the study and the clear presented data make me consider the paper can be published in the present form.

Additional comments:

1. What is the main question addressed by the research?
2. Do you consider the topic original or relevant in the field, and if
so, why?
3. What does it add to the subject area compared with other published
material?
4. What specific improvements could the authors consider regarding the
methodology?
5. Are the conclusions consistent with the evidence and arguments
presented and do they address the main question posed?
6. Are the references appropriate?
7. Please include any additional comments on the tables and figures.

1.Starting from the problem of the real relevance of bone targeted treatments to prevent metastasis, (bisphosphonates and denosumab), the author presents a documented analysis of their role especially in breast cancer.

2.The topic is relevant in the field, as long as the problem of the role of adjuvant bisphosphonates therapy according to the stage and type of cancer is still in discussion.

3. The paper summarizes the results of  all the important relevant recent clinical trials in the field and the conclusions may offer a better understanding of bone targeting agents actions.

4.Maybe a more concise and more methodically evaluation of bone targeting agents impact on preventing metastasis.

5.The conclusions are consistent with evidence;  in the same time the author is pointing out  the role of other new approaches in deciphering and resolving the problem of recurrences in breast cancers.

6.All the referrences are connected to the subject of this paper.

7.The MAF test(Table 1)   is properly used to determine the correlation between biosphosphonates administration and the outcome of patients

Author Response

Thank you for the positive comments. No specific changes requested

Reviewer 2 Report

Prof. Coleman presents an interesting overview on the clinical use of bone-targeted drugs in an adjuvant setting for breast cancer, and other bone-metastasising tumours. Although there is no doubt that the review is well-written and no one is more qualified than Prof. Coleman to write about this topic, there are some minor observations that I think will further improve the report, as detailed below.

General comments: I think it would be important to give some more details about the possible therapeutic options, and how they work in bone. For example the mechanism of action of denosumab and bisphosphonates. Additionally, clinical guidelines are certainly important to have, but discussing potential future applications of other types of adjuvant therapies might make the review even more interesting to the readership. The author is very familiar with other bone-targeted drugs such as Ra233 so there is no doubt he would be able to write a very interesting discussion on the matter.

Although this review is aimed at experts in the field and medical doctor, a brief definition of "adjuvant" and "neo-adjuvant" could be included in the introduction.

A conflicts of interest statement is missing.

Specific comments:

Title and throughout the text (e.g. line 80): bone-targeted is sometimes written withouth hyphen

Line 17: it is a matter of tone, but if Prof. Coleman deems it appropriate, "demonstrable" could be substituted with "clear"

Line 20: "RR" is missing

Line 23: I think it would be good to specify that they both act mainly on osteoclasts

Line 28: please expand acronym "MAF"

Line 42: typo "multi=step"

Line 50: cellular dormancy is certainly an extremely important phenomenon in bone metastasis, but is it the norm? I am not sure whether any specific studies evaluating the incidence of dormancy in breast cancer are present, either way an estimation on the importance of the phenomenon could be presented, e.g. "it has been estimated that up to 20% of breast cancers remain dormant in bone after surgical resection and adjuvant therapy"

Line 57: Probably the author meant "is a particular feature"

Line 81: I think "influencing" should be "influence"

Line 84: it would be best to state that this will be discussed below in detail

Line 85: the title could be more specific, e.g. "adjuvant bone-targeted agents in breast cancer". Also, "breast cancer" is the only paragraph title, there is none for "prostate cancer" and "lung cancer". Of course there is far less literature for those, so a second paragraph could be titled "adjuvant bone-targeted agents in other bone-metastatic cancers"

Line 88: it would be nice to reference this first clinical trial

Line 94: practise-changing

Line 120: I think there is some problem with the parentheses

Line 162: number 73 repeated twice

Line 163: please comment on why

Line 177: formatting of the numbers in parentheses inconsistent with the rest

Line 190: please expand acronym RANK

Figure 1: it is a little "squished" top to bottom, could it be slighly expanded vertically to make it look better?

Line 231: please expand acronym PTHrP

Line 252: was this caused by the treatment?

Author Response

Reviewer 2: Prof. Coleman presents an interesting overview on the clinical use of bone-targeted drugs in an adjuvant setting for breast cancer, and other bone-metastasising tumours. Although there is no doubt that the review is well-written and no one is more qualified than Prof. Coleman to write about this topic, there are some minor observations that I think will further improve the report, as detailed below.

General comments: I think it would be important to give some more details about the possible therapeutic options, and how they work in bone. For example the mechanism of action of denosumab and bisphosphonates. Additionally, clinical guidelines are certainly important to have, but discussing potential future applications of other types of adjuvant therapies might make the review even more interesting to the readership. The author is very familiar with other bone-targeted drugs such as Ra233 so there is no doubt he would be able to write a very interesting discussion on the matter.

  • Ra223 has only been studied in advanced metastatic disease so we do not know if it is of any relevance in the adjuvant setting. Have mentioned briefly and referenced a preclinical study suggesting possible reduction in development of bone metastasis within a short new section entitled “Future perspectives”.

Although this review is aimed at experts in the field and medical doctor, a brief definition of "adjuvant" and "neo-adjuvant" could be included in the introduction.

  • Added a sentence explaining adjuvant treatments and impact on survival

A conflicts of interest statement is missing.

  • Added

Specific comments:

Title and throughout the text (e.g. line 80): bone-targeted is sometimes written withouth hyphen

  • Now with hyphen throughout

Line 17: it is a matter of tone, but if Prof. Coleman deems it appropriate, "demonstrable" could be substituted with "clear"

  • Deleted “demonstrable”

Line 20: "RR" is missing

  • Added

Line 23: I think it would be good to specify that they both act mainly on osteoclasts

  • Added as suggested

Line 28: please expand acronym "MAF"

  • Added

Line 42: typo "multi=step"

            Corrected

Line 50: cellular dormancy is certainly an extremely important phenomenon in bone metastasis, but is it the norm? I am not sure whether any specific studies evaluating the incidence of dormancy in breast cancer are present, either way an estimation on the importance of the phenomenon could be presented, e.g. "it has been estimated that up to 20% of breast cancers remain dormant in bone after surgical resection and adjuvant therapy"

  • More than 50% of relapses occur more than 5 years after diagnosis of the primary treatment so tumour dormancy is extremely important and the norm in patients with breast (and prostate) cancers. This is explained in lines 56-58 and a specific reference for late relapse in breast cancer substituted for the previous reference 3.

Line 57: Probably the author meant "is a particular feature"

  • Thank you. Corrected

Line 81: I think "influencing" should be "influence"

  • Thank you. Corrected

Line 84: it would be best to state that this will be discussed below in detail

  • Added as suggested

Line 85: the title could be more specific, e.g. "adjuvant bone-targeted agents in breast cancer". Also, "breast cancer" is the only paragraph title, there is none for "prostate cancer" and "lung cancer". Of course there is far less literature for those, so a second paragraph could be titled "adjuvant bone-targeted agents in other bone-metastatic cancers"

  • Title revised and numbering corrected as this is part of section 2
  • Title for other tumours added as requested and numbered 2.2

Line 88: it would be nice to reference this first clinical trial

  • Original clodronate trials referenced in place of Cochrane review and reference numbers amended accordingly

Line 94: practise-changing

  • Practice-changing is correct

Line 120: I think there is some problem with the parentheses

  • Corrected

Line 162: number 73 repeated twice

  • Corrected

Line 163: please comment on why

  • Since submission a further update of ABCSG18 has been presented at ASCO and published in abstract format. As a result, the section on ABCSG18 and the implications of the updated results have been revised.

Line 177: formatting of the numbers in parentheses inconsistent with the rest

  • Corrected

Line 190: please expand acronym RANK

  • Defined in lines 63/64

Figure 1: it is a little "squished" top to bottom, could it be slighly expanded vertically to make it look better?

  • Changed as suggested

Line 231: please expand acronym PTHrP

  • Defined in line 62

Line 252: was this caused by the treatment?

  • Yes, we believe so. Text clarified

Reviewer 3 Report

The manuscript is a well-written review article. Inclusion of the MAF story as a subsection of the clinical application may make it more prominent.

Author Response

Thank you for the positive review. No changes suggested